# Enhanced Autophagy in Damaged Laminar Tissue of Acute Laminitis Induced by Oligofructose Overloading in Dairy Cows

**DOI:** 10.3390/ani13152478

**Published:** 2023-07-31

**Authors:** Muhammad Abid Hayat, Jiafeng Ding, Xianhao Zhang, Tao Liu, Jiantao Zhang, Shehla Gul Bokhari, Hamid Akbar, Hongbin Wang

**Affiliations:** 1Department of Veterinary Surgery, College of Veterinary Medicine, Northeast Agricultural University, Harbin 150030, China; abid.hayat@uvas.edu.pk (M.A.H.); zhxh950717@outlook.com (X.Z.); liutaotiger@163.com (T.L.); zhangjiantao@neau.edu.cn (J.Z.); 2Heilongjiang Key Laboratory for Laboratory Animals and Comparative Medicine, Harbin 150030, China; 3College of Animal Science and Technology, Guangxi University, Nanning 530004, China; jiafengding@cau.edu.cn; 4Department of Veterinary Surgery and Pet Sciences, Faculty of Veterinary Sciences, University of Veterinary and Animal Sciences, Lahore 54000, Pakistan; shehla.gul@uvas.edu.pk (S.G.B.); hamid.akbar@uvas.edu.pk (H.A.)

**Keywords:** autophagy, acute laminitis, dairy cows, laminar tissue, oligofructose

## Abstract

**Simple Summary:**

Laminitis is a serious hoof disease in dairy cows, which causes lameness and a reduction in milk production. It causes huge economic loss to dairy industries worldwide. However, its pathogenesis remains with a limited study. Autophagy may play a role in the pathogenesis of laminitis in dairy cows. In this experiment, the autophagy status in the laminar tissue of dairy cows with oligofructose (OF)-induced laminitis was examined at the gene and protein level. Increased gene expression levels of ATG5, ATG12, and Beclin1 were observed in the OF group’s laminar tissues in dairy cows. However, gene expression levels of P62 and mTOR were decreased, while the protein expression level of Beclin1 was increased. In addition, the decreased protein expression level of LC3II and P62 were observed in the OF group’s laminar tissues in dairy cows. These results indicated that the imbalanced gene and protein level condition of autophagy-related markers may be the basic cause for the basement membrane detachment from the epidermal lamellae, which confirmed that autophagy was enhanced in the laminar tissues by OF in treated dairy cows.

**Abstract:**

This study was aimed at determining the autophagy activity in the laminar tissue of dairy cows with oligofructose-induced laminitis. Twelve healthy non-pregnant Holstein cows were randomly divided into two groups of six cows each, entitled the control group and the oligofructose overload group (OF group), respectively. At 0 h, cows in the OF group were gavaged with oligofructose (17 g/kg BW) dissolved in warm deionized water (20 mL/kg BW) through an oral rumen tube, and the dairy cows in the control group were gavaged with the same volume of deionized water by the same method. At −72 h before, as well as 0 h, 6 h, 12 h, 18 h, 24 h, 36 h, 48 h, 60 h, and 72 h after perfusion, clinical evaluations of both groups were monitored. After 72 h, the laminar tissues of the dairy cows in both groups were collected to examine the genes and proteins. The gene expression of ATG5, ATG12, and Beclin1 significantly increased (*p* < 0.05), whereas that of P62 and mTOR significantly decreased (*p* < 0.01) in the OF group relative to the control group. The protein expression of Beclin-1 significantly increased (*p* < 0.05), while that of LC3II significantly decreased (*p* < 0.05) in the OF group relative to the control group. However, the protein expression of P62 non-significantly reduced (*p* > 0.05) in the OF group comparative to the control group. Furthermore, the distribution of the Beclin1 protein in the laminar tissue significantly increased (*p* < 0.01), while that of the P62 protein significantly decreased (*p* < 0.05) in the OF group than the control group. These findings indicate that the imbalanced gene and protein-level status of autophagy-related markers may be the basic cause for the failure of the epidermal attachment. However, a more detailed gene and protein-level study is needed to further clarify the role of autophagy in the pathogenesis of bovine laminitis.

## 1. Introduction

Lameness is one of the most serious health problems in dairy cows worldwide [1], significantly affecting animal welfare and causing huge economic losses to the dairy industry. Laminitis in dairy cows is the most common hoof disease and can be defined as a diffuse, sterile inflammation in the dermal papilla and vascular layer of the hoof wall [2]. This disease may cause various laminitis-related claw horn disruptions (CHDs), such as sole ulcer, sole hemorrhage, and white line disease [3]. Furthermore, it can compromise the structural integrity of the lamellae, which consists of two interdigitated layers (the dermal and epidermal lamellae) [2]. The third phalanx (P3) is joined to the inside hoof wall by lamellae in the capsule structure. Detachment of the epidermal and dermal lamellae causes sinking and rotation of the P3 within the capsule, resulting in significant lameness and pain [4]. The basement membrane (BM) is an interface between the two layers, and an important component of the extracellular matrix (ECM). Separation and breakdown of the BM were typical histological changes primarily inducing impaired epidermal attachment leading to laminitis in dairy cows [5,6]. The underlying molecular pathway is still unknown.

In clinical cases, dairy cow laminitis is usually secondary to numerous diseases, such as mastitis, metritis, and ruminal acidosis [7]. Many of these disorders are caused by the production of lipopolysaccharide (LPS), and other toxic chemicals, such as histamine (HIS) or lactate (LA), which can also flow through the systemic blood circulation and reach the hoof, causing laminitis [8]. By simulating similar clinical conditions, experimental models have been developed for bovine laminitis. The oligofructose (OF) overload model is more widely and consistently used, rather than other inducing models [9,10]. This model may indicate the same histological and pathophysiological alterations seen clinically in cows suffering from laminitis [11,12]. A recent study [13], reported that excessive oligofructose can result in serious clinical problems, such as acute laminitis and respiratory adaptations for metabolic acidosis. Until now, the OF overload model has been extensively used in equine for laminitis investigations, compared to laminitis in bovine [14,15].

Cellular survival is based on recycling, especially during sudden physiological alterations [16]. Recycling controls the amount of intracellular components. Autophagy (Greek for “self-eating”) is a well conserved recycling process that involves the lysosome-mediated breakdown of cellular components [17]. Few studies have reported autophagy in dairy cows, despite its widespread study in vitro and in other animal models. Autophagy is essential for upholding the stability of intracellular material and the metabolism. Autophagy may contribute significantly to the progression of metabolic disorders. Recent studies have confirmed that autophagy is increased in the livers of dairy cows with clinical ketosis and fatty liver, indicating that autophagy plays a significant role in maintaining metabolic stability in the cow’s liver [18,19]. To keep cells in a state of equilibrium, autophagy is normally only minimally produced under normal physiological settings [20]. Yet, autophagy is adaptively increased when cells are exposed to external stimuli like bacteria and toxins to engulf bacteria or intracellular toxic compounds like reactive oxygen species, hence maintaining cell homeostasis [21]. Nonetheless, irregular autophagy leads to the accumulation of autophagic vesicles that lead to cell death [22]. According to recent research by [23], subacute ruminal acidosis (SARA) promotes autophagy in the livers of dairy cows given a high-concentrate diet. It is believed that autophagy in hepatocytes is adaptively increased to deal with the oxidative damage to the liver caused by SARA. Regrettably, no reports have been made on studies on autophagy in the laminar tissue of laminitis cows and equine laminitis.

To our knowledge, no study on the autophagy status of laminar tissue in dairy cows with oligofructose-induced laminitis has been conducted. Therefore, the current investigation was primarily based on the determination of the laminar tissue autophagy status in dairy cows with oligofructose-induced laminitis at the gene and protein level, and on the provision of significant therapeutic targets for this disease in the future. We hypothesized that oral oligofructose challenge may induce autophagy in the laminar tissue of laminitic dairy cow hooves.

## 2. Materials and Methods

### 2.1. Experimental Animals

Twelve clinically sound, non-pregnant Chinese Holstein dairy cows [24], excluding those with a history of serious hoof lesions, 18–26 months old (20.67 ± 3.01 mo), weighing 335–403 kg (379.71 ± 19.87 kg), and with a body condition score (BCS) [25] ranging from 2.7–3.3 (3.00 ± 0.23) were selected from Qingxi dairy farm in the Xiangfang District of Harbin, P.R. China. The average age in months, or body condition score and body weight, between the two groups of dairy cows had no significant difference. The cows were reared in the large animal laboratory in Northeast Agricultural University’s Animal Hospital. The ground was rubberized, and the cows were free to drink water and eat hay grass ad libitum. The surroundings were kept clean and sanitary. During the thirty day acclimation period, the cows’ hooves were trimmed in advance, and the cows were led on a walk every day and trained to accept physiological parameters and related scoring checks. After the acclimation period ended, the experimental animals had shown no signs of discomfort before the experiment began.

### 2.2. Experimental Design and Treatment

A total of 12 experimental dairy cows were randomly divided into 2 groups of 6 individuals: an OF overload group and a control group. According to the body weight of the cows in the OF-treated group, the low-dose solution (0.85 g/kg of oligofructose (Shandong Shenglong Technology Co., Ltd., Jinan, China) dissolved in 0.1 L/100 kg of warm deionized water) and the high-dose solution (17 g/kg of oligofructose dissolved in 2 L/100 kg of warm deionized water) were prepared. Three days before the high oral dose of the OF aqueous solution (−72 h) was administered, the OF-treated cows were given a low-dose OF solution (0.85 g/kg OF dissolved in 0.1 L/100 kg of warm deionized water), while 0.1 L/100 kg of warm deionized water was given to the control cows via a stomach tube. On the day the high oral dose of the OF aqueous solution (0 h) was administered, the OF-treated cows received 17 g/kg BW of oligofructose dissolved in 20 mL/kg BW of warm deionized water, while the control cows were administered with 20 mL/kg BW of warm deionized water at 0 h (on the day of the high oral dose) via a stomach tube, following the method described by [9,10].

Considering animal welfare, supportive treatments, such as 1.96 mL/kg BW of calcium borogluconate (19.6 mg of Ca/mL; Heping Animal Medicine Co., Ltd., Harbin, China) were given at 18 h, and 15 mL/kg BW of Ringer lactate (Heping Animal Medicine Co., Ltd., Harbin, China) and 126 mg/kg BW of sodium bicarbonate (Heping Animal Medicine Co., Ltd., Harbin, China) were given at 18 h and 24 h, following administration of the OF to the dairy cows.

### 2.3. Clinical Examination of Dairy Cows

The cows in the control group and OF-treated group were recommended for clinical examination comprising of the respiration rate, heart frequency, rectal temperature, hoof temperature, hoof shell temperature, rumen contraction rate, rumen pH (Benchtop pH meter, Mettler Toledo Inc., Greifensee, Switzerland), eating and drinking routine, feces consistency, weight shift, wrist (tarsal) joint swelling score, hoof crown band swelling score, digital artery pulsation score, blood pressure (electronic sphygmomanometer), hoof pain score, and claudication score at −72 h, 0 h, 6 h, 12 h, 18 h, 24 h, 36 h, 48 h, 60 h, and 72 h [9,10]. During the claudication stage, the cows were allowed to walk and trot by traction in a straight line and then to try a circle movement on the same ground at the Animal Hospital, Northeast Agricultural University Harbin, P.R. China. The claudication scores for each cow were evaluated by three expert licensed veterinarians, according to the Sprecher scoring method [24]. When all the expert veterinarians evaluated a score of ≥2 for a cow, then it was inspected as lame. According to the Sprecher method, the claudication scoring was graded as 1 = normal, grade 2 = mildly lame, grade 3 = moderately lame, grade 4 = lame, and grade 5 = severely lame. After 72 h of the OF overload, a dose of 20 mg/kg of phenytoin sodium and pentobarbital sodium (fatal plus; 20 mg/kg IV) were administered intravenously to all the animals to be euthanized [9,10]. 

### 2.4. Laminar Tissue Sampling

After 72 h of oligofructose overload, when the clinical observations of the cows in the treated group met the criteria for acute laminitis in dairy cows [9,10], the cows in the control and treatment groups were euthanized, and hoof laminar tissue samples were collected. According to the method explained in [11], the left hind hoof of the diseased animal was isolated within 5–10 min, placed in an ice box, and brought back to the laboratory as soon as possible. The cow’s hoof wall was sectioned by using a band saw to uncover the laminar tissue, which was then isolated from the hoof shell. The laminar tissue was sectioned into tiny tissue pieces (1–2 mm^2^), followed by being frosted in liquid nitrogen and stored at −80 °C. The residual hoof tissue was embedded in a 10% formalin-phosphate buffer and fixed overnight (room temperature). The whole procedure was conducted on ice. The operators wore sterile gloves and disposable masks to avoid tissue contamination. Usually, the laminar wall runs along the entire axial, abaxial, and dorsal sides of the hoof, measuring 3–7 cm from the coronet area to the sole corium. The isolated laminar tissue wall follows 2 cm under the coronet and contains the axial and abaxial parts of the laminar wall tissue. The original laminar tissue measured 6 cm in length and 3 cm in width. 

### 2.5. RNA Isolation and cDNA Synthesis

Following the manufacturer’s instructions, the total RNA was collected from the laminar tissue of 12 cows’ hooves utilizing the RNA Miniprep Kit (Invitrogen, Carlsbad, CA, USA). The laminar tissue samples (100 mg) were homogenized with 1 mL of TRIzol reagent (Invitrogen, Carlsbad, CA, USA). The entire RNA without DNA, protein, and isopropanol precipitates were extracted with chloroform and rinsed with 75% ethanol in a non-DNAse/RNAse centrifuge tube. Using an ultra-nucleic acid protein detection kit (NO-ONE Gene, USA), the purity and concentration of the extracted total RNA were assessed. The integrity of each RNA sample was confirmed by 1% agarose gel electrophoresis (Bio-Rad Laboratories, Hercules, CA, USA). The collected RNA tissue samples were diluted to 1 µg/µL by quantifying optical density. The total RNA (1 µg) was collected from each tissue sample following the manufacturer’s instructions for Prime-Script™ RT Kit (Takara, Dalian, China). Reverse transcription was performed to obtain the cDNA. The complementary DNA (cDNA) was diluted 1:3; diluted with DEPC water and stored at −20 °C for RT-qPCR analysis. Diethylpyrocarbonate (DEPC) water is formed from dH2O with DEPC and, then, autoclaved to eliminate the DEPC.

### 2.6. Quantitative Real-Time Polymerase Chain Reaction (RT-qPCR)

The primers utilized in this study for detecting the genes, including glyceraldehyde-3-phosphate dehydrogenase (GAPDH), autophagy related 5 (ATG5), autophagy related 12 (ATG12), autophagy related 6 (ATG6/Beclin-1), Sequestosome 1 (P62/SQSTM1), and the mammalian target of rapamycin (mTOR) were designed by Shanghai Sheng Gong Biotechnology, Co., Ltd. (BBI Life Sciences, Shanghai, China) (Table 1). The specificity of each primer sequence was determined by the Blast computer program from (NCBI) the National Center for Biotechnology Information database (http://blast.ncbi.nlm.nih.gov/, accessed on 22 June 2023).

The RT-qPCR was executed by the green chimeric florescence detection technique using the SYBR Premix Ex Taq TM ‖ Kit (Takara, Dalian, China) with the LightCycler 480 RT-PCR system (Roche, Mannheim, Germany). The (20 µL) PCR reaction mixture contained a (2 µL) cDNA template and (18 µL) PCR major blend. The PCR major blend comprised (6.4 µL) DEPC water, (10 µL) SYBR green florescence dye, and (1.6 µL) miscellaneous primer solution (10 µM for each forward and reverse primer). The ultimate primer concentration was 0.4 µM/µL. The following PCR settings were used: pre-denaturation, 1 cycle, 95 °C, 30 s; quantitative analysis, 40 cycles of 95 °C 5 s and 60 °C 30 s; melting curve analysis, 95 °C 5 s, 65 °C 15 s, 95 °C 5 s 1 cycle; cycling, then cooling, 1 cycle at 50 °C for 30 s. The Ct values for each gene were assessed by the Light Cycler 480 software 2.0 (Roche, Mannheim, Germany) and the Abs Quant/Fit Points method. According to the ΔΔCt method, with GAPDH as the housekeeping gene, the PCR efficiency of each gene was analyzed.

### 2.7. Western Blot

The hoof laminar tissue sample (100 mg) was standardized with 1 mL radioimmunoprecipitation assay (RIPA) lysate buffer (Beyotime Biotechnology, Shanghai, China). The RIPA lysate buffer contained 0.1% SDS, 1% deoxycholate, and 1% Triton-X-100. During homogenization, the tissue was treated with 10 µL of the protease inhibitor PMSF (Beyotime Biotechnology, Shanghai, China) and, then, the sample was processed in a grinder at 4 °C for 4 min. The sample was then centrifuged at 12,000 rcf/min for 15 min. The BCA method (Beyotime Biotechnology, Shanghai, China) was utilized to calculate the total protein concentration. The amount of protein in the tissue sample was determined (2.5 µg/µL) by sodium dodecyl sulphate (SDS-PAGE) protein loading buffer and physiological saline in boiled water for 5 min. Each well received a 9 µL protein sample. Following separation on a 10% polyacrylamide gel, the total amount of protein (25 µg) was shifted to a nitrocellulose (NC) filter membrane (Pall Life Sciences Inc., Pensacola, FL, USA) using the semi-dry method (300 mA, 1.5 h). For blocking, the membrane was placed in 5% skimmed milk in TBS (Tris-buffer saline; Leagene Biotechnology, Beijing, China) with Tween-20 (0.1% Tween-20 TBS, TBST; Leagene Biotechnology, Beijing, China) for 2 h at ambient temperature with rocking. After blocking, the membranes were then coated with the primary antibody overnight at 4 °C. The specific primary antibodies with dilutions were utilized as Beclin-1 (1:1000), P62 (1:1000), and microtubule-associated protein light chain 3-II (LC3-II) (1:1000), and β-actin at 1:2000 dilution (BIOSS Antibodies Beijing Biosynthesis Biotechnology Co., Ltd., Beijing, China). Subsequently washing with TBST three times (15 min each), the NC membranes were transferred to a diluted secondary antibody and shaken for 2 h at room temperature for incubation. The HRP-conjugated goat anti-rabbit IgG secondary antibody (dilution 1:5000) in TBST buffer was used (BIOSS Antibodies Biosynthesis Biotechnology Co., Ltd., Beijing, China). After washing with TBST three times (15 min each), the protein bands in the NC membranes were seen using a Western blot ECL substrate (Meilun Biotechnology Co., Ltd., Dalian, China) luminescent liquid and measured by densitometry using ImageJ software version 1.53t. The internal reference protein β-actin (1:2000 dilution; BBI Life Sciences, Shanghai, China) was used.

### 2.8. Immunohistochemistry

Immunohistochemistry was used to identify Beclin-1 and P62 expression in the laminar tissue. The laminar tissues samples were sliced into suitable sizes, put in 4 percent paraformaldehyde for 24 h, then incorporated and sectioned. The slices were dewaxed in an 80 °C oven overnight, and then submerged in a 3 percent hydrogen peroxide solution for 10 min in a darkened room for intrinsic peroxidase blockage, accompanied by antigen restoration in a pressurized cooker utilizing a sodium citrate antigen restoration solution. The samples were coated with BSA for 20 min at ambient temperature, subsequently incubated with the primary antibodies (1:200, Becilin-1; 1:200, P62; Novus Biologicals, Littleton, CO, USA) for a further 24 h at 4 °C, and then exposed to streptavidin-labeled horseradish peroxidase (HRP) for an additional 30 min at room temperature. After that, each section was 3,3′-Diaminobenzidine (DAB) and hematoxylin stained, fixed with mild glue, and oven dried. To determine the staining level of the tissue slices in each group, the stained samples were inspected under a microscope and assessed with Image-Pro Plus version 6.0 software-IPWIN16.EXE (Media Cybernetics Image-Pro Software, USA).

### 2.9. Statistical Analysis 

The data were analyzed using GraphPad Prism version 7 software (Version 7.04, Graph Pad Software Inc., San Diego, CA, USA). The experimental data were analyzed with normal distribution. All the data were evaluated using an independent Student *t* test and Bonferroni’s multiple comparison tests with a significance level of 5%. Experimental data are indicated as mean ± SD.

## 3. Results

### 3.1. Clinical Manifestation of Dairy Cow Laminitis

All the OF group dairy cows presented clinical manifestations of unique acute ruminal and systemic acidosis symptoms, such as severe watery diarrhea, in-appetence lack of food intake, anorexia, depression, swelling of the carpal (tarsal) joints, swollen hoof coronary band, change in weight shift, increased heart rate, increased diastolic blood pressure, elevated hoof temperature, increased body temperature, decreased breathing rate, hyperactivity of the digital (toe) arteries, slowed rumen contraction, hoof pain, decreased rumen pH, transient fever, and lameness [26,27,28]. The control cows indicated no signs of systemic disease. All these symptoms were the same as those reported by [9,10,11]. During the claudication assessment, clinical signs of laminitis were initially observed at 24 h after the OF overload and continued to increase until 60 h, at which point this study found a maximal limp score of 3–5, which confirmed acute laminitis. The claudication scores were constant from 60 h to 72 h, and this study did not find any further significant changes (Table 2). Therefore, the animals were then euthanized for further experimental requirements. All these symptoms were the same as those reported by Danscher et al. [10], who observed acute laminitis at 60–120 h of OF overload. 

### 3.2. Autophagy-Associated Genes Expression in Laminar Tissue of Laminitis Dairy Cows 

The autophagy-associated genes expressions, including Beclin-1, ATG5, ATG12, P62, and mTOR in the control and OF groups are indicated in Figure 1. The mRNA expression level of Beclin-1 (*p* = 0.005) was significantly higher in the laminar tissue of the OF group than in the control group (*p* < 0.01). Similarly, the ATG5 (*p* = 0.027) and ATG12 (*p* = 0.013) mRNA expressions in the sick animals increased significantly (*p* < 0.05) in the OF group’s laminar tissue than in the control group. In this study, the mRNA expression of P62 (*p* = 0.001) and mTOR (*p* = 0.001) significantly decreased (*p* < 0.01) in the sick cows than in the control cows. These findings indicated that the increased or decreased expression of genes may be linked to autophagy in sick animals confirming that autophagy was induced in the laminar tissue of laminitic dairy cows.

### 3.3. Autophagy-Associated Protein Expression in Laminar Tissue of Laminitis Dairy Cows

Autophagy-associated protein expression, such as Beclin-1, P62, and LC3II, in the OF and control groups are indicated in Figure 2. The expression of the Beclin-1 protein (*p* = 0.0374) increased significantly (*p* < 0.05) in the OF group’s laminar tissue as compared to the control group. The LC3-II protein expression (*p* = 0.0148) decreased significantly (*p* < 0.05), while the P62 protein expression (*p* = 0.545) decreased non-significantly (*p* > 0.05) in the OF group’s laminar tissue rather than in the control group. These findings indicated that increased or decreased expression of the proteins may be linked to autophagy in sick animals. The detailed Western blot pictorial representation for Beclin-1, P62, and LC3-II is provided in the Appendix A.

### 3.4. Immuno-Expression of Beclin1 and P62 Proteins in Laminar Tissue of Laminitis Dairy Cows

The immunohistochemical results for the Beclin-1 protein expression in the laminar tissue of cows indicated that the mean Beclin-1 staining in the cytoplasm of the laminar tissue of the OF group was 6.67%, while it was 2% in the control tissue. Statistically, the expression of the Beclin-1 protein (*p* = 0.0017) was increased significantly (*p* < 0.01) in the cytoplasm of the OF group’s laminar tissue as compared to the control group, as illustrated in Figure 3.

The immunohistochemical results for the P62 protein expression in the laminar tissue of the cows indicated that the mean P62 staining in the cytoplasm of the laminar tissue of the OF-treated group was 2%, while it was 5.66% in the control tissue. Statistically, the expression of the P62 protein decreased significantly (*p* < 0.05) in the cytoplasm of the OF group’s laminar tissue as compared to the control group (*p* = 0.0137), as illustrated in Figure 4.

## 4. Discussion

Oligosaccharide (OF) overload in dairy cows has been extensively studied in acute rumen acidosis (ARA), laminitis, and synovitis [10,29,30]. Excessive intake of fermentable carbohydrates leads to changes in the microbiota and the rate of fermentation. The content of lactate was increased in the rumen fluid and absorbed into the systemic circulation during acute rumen acidosis [31]. In addition to lactate, lipopolysaccharide (LPS) is also considered to be a major factor in the development of rumen acidosis. LPS is generally supposed to translocate into the blood stream during experimentally induced subacute ruminal acidosis (SARA) [32,33,34]. These substances cause systemic inflammatory responses, but the autophagy activity in the laminar tissue of dairy cows remains unclear in OF-induced laminitis. In the previous published research articles, we have demonstrated that bovine laminitis has been established effectively utilizing the OF-induced method, and OF-treated cows showed clinical characteristics of distinct acute ruminal and systemic acidosis manifestations, as well as recognizable histological signs in the laminar tissue, such as BM damage and separation, the sagging of the epidermal lamellae, along with modifications to the basal cell shape [26,27,28]. We used a similar sample of laminar tissue to the prior work and assessed the autophagy markers at the gene and protein levels.

In this study, we examined autophagy-associated gene expression, including ATG5, Beclin1, ATG12, P62, and mTOR; Beclin1, P62, and LC3II protein expression; and the immunohistochemical expression of the P62 and Beclin-1 proteins; in the laminar tissue to investigate the changes in autophagy activity in the lamellae of cow hooves with OF overload-induced laminitis. This study observed a rise in Beclin-1, ATG5, and ATG12 gene expression and a reduction in P62 and mTOR gene expression, a reduction in the protein expression of P62 and LC3II, and increased protein expression of Beclin-1 within the laminar tissues with OF overload-induced bovine laminitis. However, the results from the immunohistochemistry analysis of P62 and Beclin-1 showed that the quantity of positive cells with Beclin-1 was increased, while the quantity of P62 positive cells decreased in the laminar tissue of laminitic cows. Similar observations were reported in the livers of dairy cattle with SARA [23], mildly obese livers [19], and the mammary glands of dairy cows with hyperketonemia [35]. This demonstrably showed that autophagy overexpression was induced in the laminar tissue of OF-induced dairy cow laminitis.

Autophagy is a mechanism that is maintained by the intracellular breakdown system, by which amino acids are reused within cells, and plays numerous physiological roles for instance in proliferation, differentiation, and the upkeep of cellular homeostasis [36]. This process is regarded as involving the development of autophagosomes, double-membrane vesicles that sequester the products that are cytoplasmic in the lysosome [22]. But, unregulated autophagy can result in the building of autophagic vacuoles that can lead to cellular death [22]. Major proteins involved in a system that controls autophagy in cells of all the organisms, from yeast to humans, are produced by genes in the ATG (autophagy-related) family [37,38]. The discovery of these proteins allowed researchers to identify signalling systems that contribute to the activation and control of this process. For the key biochemical markers, two proteins, Beclin1 (Atg6), and LC3 (Atg8), as well as ATG7, are thought to be involved in autophagy. Becilin-1 is one of the major autophagy parameters in mammals and has been implicated in mammalian autophagy [39]. Beclin-1 binds PI3KC3 (Vps34) to create complexes with proteins [40] that regulate intracellular transport and the formation of autophagosomes [41]. The formation of autophagosome membranes is activated by the creation of the Becilin-1 complex. Beclin-1 also modulates activity, which is certainly autophagic [41,42]. Therefore, Beclin1 contributes to the growth of lipid bilayer membranes and the enlistment of other autophagy-related proteins. In particular, LC3II, ATG5, and ATG12 tend to be enlisted and provide stimulation for the development of autophagosomes.

The activity of autophagy mandates two ubiquitin-like reformation systems consisting of the ATG8 (LC3) reformation system and the ATG12–ATG5 reformation system. LC3 has been identified as the only mammalian protein with stable autophagosome membrane binding. ATG4 quickly shears the carboxyl terminus of the newly synthesized LC3 protein, resulting in the cytosolically localized LC3I. During ubiquitination processing, LC3I is changed to bind to PE located on the autophagosome membrane, resulting in membrane bound LC3II that is confined within the autophagosome. The protein LC3II, which may be identified in autophagosomes and whose quantity is directly connected to the extent of the autophagy, is one of the protein markers of autophagy. In response to ATG10, ATG7 activates ATG12, which then attaches to ATG5 to create the ATG12–ATG5 complex. The ATG5–ATG12 complex, acting as an E3 enzyme, stimulates the production of ATG8-PE during ubiquitin-binding processes. For autophagosome formation to occur, the ATG8-PE and ATG16–ATG5–ATG12 complexes must first be formed [43]. P62 serves as an autophagy feedback regulator by encasing and recognizing degraded substrates in the autophagosome, eventually declining as autophagy increases [44,45]. The versatile protein P62, also known as SQSTM1, is essential for signaling and selective autophagy. Specifically, P62 is an adaptor protein that interacts with LC3-II to direct ubiquitinated cargos toward autophagy-specific destruction [46]. Under typical physiological conditions, the regulator known as mTOR both activates and suppresses autophagy. The stimulation of mTOR may restrict the regulation of autophagy. When metabolic stress occurs, mTOR expression is downregulated, its activity is suppressed, and autophagy is triggered [47].

## 5. Conclusions

In conclusion, the obtained results from the current study demonstrate that the imbalanced gene and protein-level condition of autophagy-related parameters may be the root cause for the failure of epidermal attachment. This study indicated that autophagy may play a role in the pathogenesis of acute laminitis in dairy cows, which directly or indirectly leads to the damage of laminar tissue in dairy cows. This result may provide a theoretical basis and fundamental knowledge for the prevention and management of laminitis in dairy cows. However, a detailed gene and protein-level investigation is needed to clarify the role of autophagy in the pathogenesis of laminitis in dairy cows.

## Figures and Tables

**Figure 1 animals-13-02478-f001:**
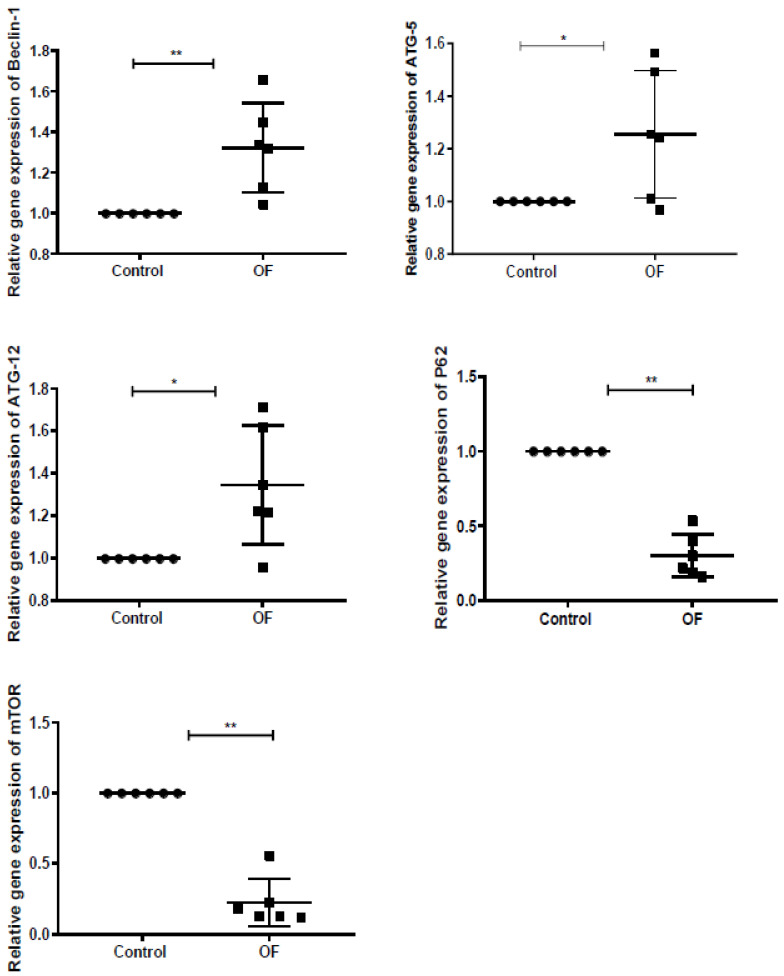
Quantitative Real-Time Polymerase Chain Reaction (RT-qPCR) results on the mRNA concentration of autophagy-related genes in the laminar tissue of both groups. Where the abbreviations are as follows: autophagy related 5 (ATG5), autophagy related 12 (ATG12), autophagy related 6 (ATG6/Beclin-1), Sequestosome 1 (P62/SQSTM1), the mammalian target of rapamycin (mTOR), and OF indicates oligofructose. “*” indicates (*p* < 0.05); “**” indicates (*p* < 0.01).

**Figure 2 animals-13-02478-f002:**
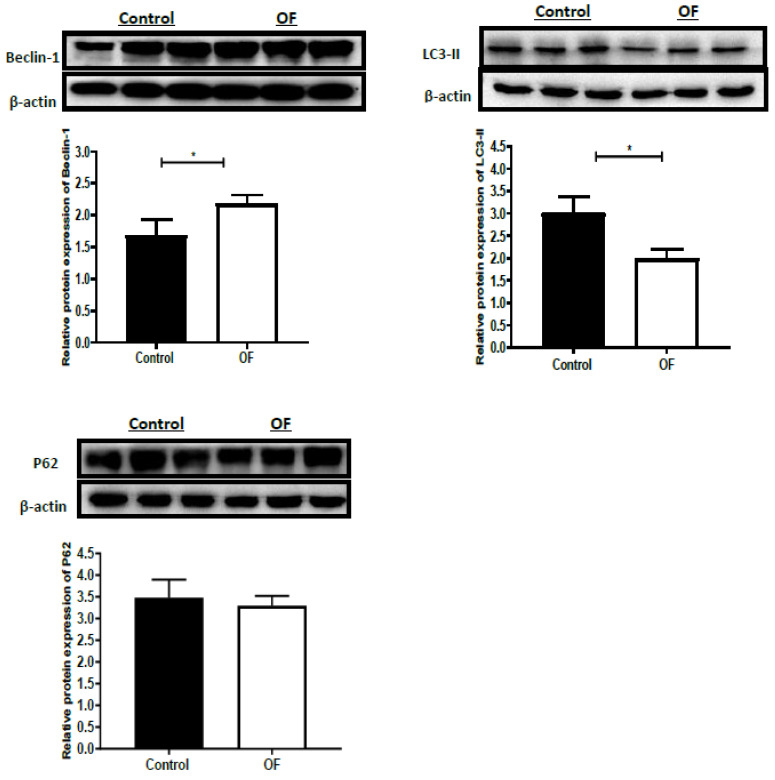
Western blot results for the autophagy-related protein expression in the laminar tissue of both groups, including autophagy related 6 (ATG6/Beclin-1), Sequestosome 1 (P62/SQSTM1), and microtubule-associated protein light chain 3-II (LC3-II). OF indicates oligofructose. “*” indicates (*p* < 0.05).

**Figure 3 animals-13-02478-f003:**
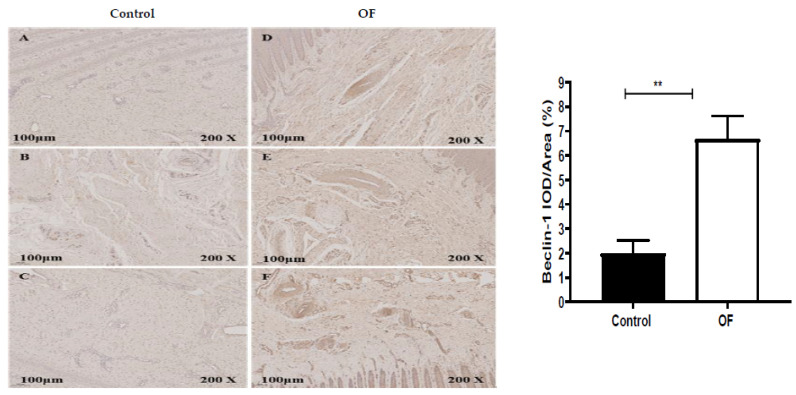
Immunohistochemical staining of autophagy related 6 (ATG6/Beclin-1) in laminar tissues: scale = 100 μm, 200×; (**A**–**C**) control cows; (**D**–**F**) OF (oligofructose)-treated cows. “**” indicates (*p* < 0.01).

**Figure 4 animals-13-02478-f004:**
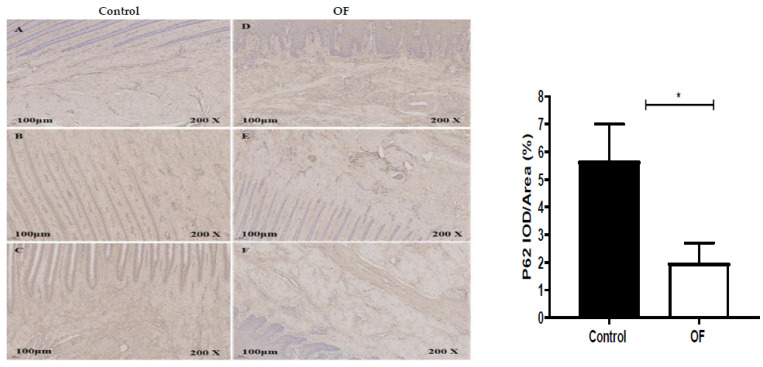
Immunohistochemical staining of Sequestosome 1 (P62/SQSTM1) in laminar tissues: scale = 100 μm, 200×; (**A**–**C**) control cows; (**D**–**F**) OF (oligofructose)-treated cows. “*” indicates (*p* < 0.05).

**Table 1 animals-13-02478-t001:** Primer sequences.

Genes	RefSeq Accession No.	Primer Sequences (5′-3′)
ATG-5	NM_001034579.2	Forward: ACACCTTTGCAGTGGCTGAGTG Reverse: TCTGTTGGTTGCGGGATGATGC
ATG-12	NM_001076982.1	Forward: GAGCGAACCCGAACCATCCAAG Reverse: AGGGTCCCAACTTCCTGGTCTG
Beclin-1	NM_001033627.2	Forward: GCAGGTGAGCTTCGTGTGTCAG Reverse: GCTGGGCTGTGGCAAGTAATGG
mTOR	XM_002694043.6	Forward: GCACGTCAGCACCATCAACCTC Reverse: CAGCCGCCGCAGCCATTC
P-62	NM_176641.1	Forward: CCAGGAGGTGCCCAGAAACATG Reverse: AGGCGGAGCATAGGTCGTAGTC
GAPDH	DQ402990	Forward: GGGTCATAAGTCCCTCCACGA Reverse: GGTCATAAGTCCCTCCACGA

**Table 2 animals-13-02478-t002:** Results of the claudication score.

Time	Control Cows (*n* = 6)						OF-Treated Cows (*n* = 6)					
	1	2	3	4	5	6	1	2	3	4	5	6
−72 h	1	1	1	1	1	1	1	1	1	1	1	1
0 h	1	1	1	1	1	1	1	1	1	1	1	1
6 h	1	1	1	1	1	1	1	1	1	2	1	1
12 h	1	1	1	1	1	1	2	1	1	2	1	1
18 h	1	1	1	1	1	1	2	2	1	3	2	2
24 h	1	1	1	1	1	1	2	2	2	3	3	3
36 h	1	1	1	1	1	1	2	3	2	3	3	3
48 h	1	1	1	1	1	1	3	3	2	3	3	3
60 h	1	1	1	1	1	1	4	4	3	5	4	5
72 h	1	1	1	1	1	1	4	4	3	5	4	5

## Data Availability

All the data generated or analyzed during the study are included in this published article.

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
