# Peer review of "Enhanced Autophagy in Damaged Laminar Tissue of Acute Laminitis Induced by Oligofructose Overloading in Dairy Cows"

_animals, 2023, doi:10.3390/ani13152478_

Round 1

Reviewer 1 Report

Dear authors,

As a veterinarian and lameness expert in Europe I have read your manuscript with exessive interest. Your study and paper is only about acute laminitis which is not a really big issue in Western-Europe. This does not mean that your research is not valuable, but accept your limitations and about your conclusion< I suggest that you limit that to : autophagy may play a role in the pathogenesis of acute laminitis

other points:

line 42: Non-infectious claw disorders, presenting as sterile...

line 43: claw disorders..

line 44-49: reconsider that part, whereby AL is compared with Claw-Horn disruption and SH and SU is more related to CHD. This part is not in line with the current accepted opinion about non-infectious claw disorders.

line 57: also prevention of course

remaining: do not use abbreviations without explaining at the first use: e.g. LPS, HIS, LA, also in M&M: DEPC water, GAPDF, mTor etc. 

line 98: strange that ref. for clinically sound etc cows.

line 138: claudication score, what were criteria and how performed??

line 153: explained by Thoefner et al. [11], 

line 186-187: abbreviation after the name 

line 250: I cannot imagine that all cows had all symptoms and we need data in a table about the different symptoms e.g. was temp. 39.2 or 40.0 or at least whatI want is mean and SD of data here 

you only present here results and no ref. to other studies, we do that normally in discussion!!

Table 2 is not referred to in the text

line 269-275: pay attention to the no. of figures behind the dot Max 2. with this no. of animals

3.3 and 3.4 the same and line 331, you can compare 6.67 wit 2% etc. etc. 

In tables and figures explain again and again your abbreviations 

line 370: septic pleuropneumonia in dairy cattle, I hardly diagnosed here, mastitis is more logical here.

line 382: previous publications? which one: we as readers want to know where we can find

line 448: shows a role is to strong: may play a role 

Author Response

Replies to the reviewer’s comments:

Reviewer-1

Comment: As a veterinarian and lameness expert in Europe I have read your manuscript with exessive interest. Your study and paper is only about acute laminitis which is not a really big issue in Western-Europe. This does not mean that your research is not valuable, but accept your limitations and about your conclusion< I suggest that you limit that to : autophagy may play a role in the pathogenesis of acute laminitis.

Response: Thank you very much for your comments and suggestion on our manuscript. These comments are very helpful for revising and improving our paper. According to the reviewer detailed suggestions, we have made a careful revision on the main manuscript (autophagy may play a role in the pathogenesis of acute laminitis in dairy cows).

Comment: line 42: Non-infectious claw disorders, presenting as sterile...;  line 43: claw disorders..; line 44-49: reconsider that part, whereby AL is compared with Claw-Horn disruption and SH and SU is more related to CHD. This part is not in line with the current accepted opinion about non-infectious claw disorders. line 57: also prevention of course.

Response: Respected reviewer, yes, non-infectious claw disorder presenting a sterile and we have revised that part in the introduction section on the main manuscript.

Comment: remaining: do not use abbreviations without explaining at the first use: e.g. LPS, HIS, LA, also in M&M: DEPC water, GAPDF, mTor etc.

Response: Respected reviewer, thank you for your suggestion, abbreviation after the name has been corrected and incorporated on the main manuscript.

Comment: line 98: strange that ref. for clinically sound etc cows.

Response: Respected reviewer, for clinically sound cows, we have linked with reference that is related to lameness/claudication scoring scale described by Sprecher et al.

Comment: line 138: claudication score, what were criteria and how performed??

Response: Respected reviewer, During claudication stage, cows were allowed to walk and trot by traction in a straight line and then to try circle movement on the same ground at the Animal Hospital, Northeast Agricultural University Harbin, P.R. China. The claudication/lameness scores of each cow were evaluated by the three expert licensed veterinarians according to the Sprecher scoring method [24]. When all the expert veterinarians evaluated a score of ≥ 2 in a cow, then it was inspected as lame. According to Sprecher method, the claudication scoring was graded as 0 = normal, grade 1 = mildly lame, grade 2 = moderately lame, grade 4 = lame and grade 5 = severely lame.

Comment: line 153: explained by Thoefner et al. [11],

Response: Yes, it was explained by Thoefner et al. [11]. The procedure for sampling of hoof laminar tissue was explained.

Comment: line 186-187: abbreviation after the name.

Response: As reviewer’s suggestion, abbreviation after the name has been corrected and incorporated on the manuscript.

Comment: line 250: I cannot imagine that all cows had all symptoms and we need data in a table about the different symptoms e.g. was temp. 39.2 or 40.0 or at least whatI want is mean and SD of data here. you only present here results and no ref. to other studies, we do that normally in discussion!!

Response: Respected reviewer, This model was previously established (ie,https://doi.org/10.3389/fvets.2020.00351), (https://doi: 10.3389/fvets.2020.597827), and (doi: dx.doi.org/10.21521/mw.6398) using the same experimental groups, this study (Autophagy status in laminar tissue) was my part of study in a same research group of that model. In the same research group, first author of this study participated equally in all data collection and treatments. The clinical examination parameters are described in our previous published articles of same research model experimental groups.

Comment: Table 2 is not referred to in the text; line 269-275: pay attention to the no. of figures behind the dot Max 2. with this no. of animals. In tables and figures explain again and again your abbreviations.

Response: Respected reviewer, Table 2 now has been referred in the text; figure and abbreviations issue has been corrected on the main manuscript.

Comment: 3.3 and 3.4 the same and line 331, you can compare 6.67 wit 2% etc. etc.

Response: Respected reviewer, in 3.3 part, for western blot images, we used ImageJ-software for collecting the data for both groups and then statistically analyzed. In 3.4 and line 331, for immunohistochemistry protein expression, the number of positive cell in laminar tissue with specific protein like Beclin1 and P62 were counted in percentage for each group using Image-Pro Plus 6.0 software (Media Cybernetics, USA). Then data was analyzed.

Comment: line 370: septic pleuropneumonia in dairy cattle, I hardly diagnosed here, mastitis is more logical here.

Response: Yes, you are right, mastitis with septic pleuropneumonia may occur. However, bovine laminitis is secondary to these diseases (septic pleuropneumonia, metritis, mastitis, ruminal acidosis etc). so in the manuscript, we just used specific diseases that laminitis is secondary to these diseases.

Comment: line 382: previous publications? which one: we as readers want to know where we can find.

Response: you can find by using the doi. (ie,https://doi.org/10.3389/fvets.2020.00351), (https://doi: 10.3389/fvets.2020.597827), and (doi: dx.doi.org/10.21521/mw.6398). In our manuscript, reference 26-28. You can find.

Comment: line 448: shows a role is to strong: may play a role.

Response: Respected reviewer, thank you for your suggestion, “ may play a role” has been incorporated  on the main manuscript.

Reviewer 2 Report

Comments to the Author

According to the research article, "Enhanced Autophagy Activity in Laminar Tissue Damage of Acute Laminitis in Dairy Cows Induced by Oligofructose Overload." One of the most common causes of dairy cow disease is laminitis. Even though the reviewer appreciates the authors' efforts in achieving the current experience, he nevertheless believes that the manuscript has several errors as following.

1. ABSTRACT: The author has to be improved and this section could be concise.

2. Introduction part, this research will contribute to the compilation of scientific information on the measuring of Autophagy activity in laminar tissue of laminitis in dairy cows. The introduction section was almost thoroughly re-concisely detailed by focused on laminitis in dairy cows.

3. Material and methods part, the experiment was conducted to evaluate damage of laminar tissue.

- I am concerned about the lack of information on the collection laminar tissue previous induce by oligofructose.

- Why does the author use 17g/kg OF to define acute in dairy cows when neither the ruminal pH nor the referee method was measured?

- Why didn't the author use an animal without OF as a negative control? 

4. Discussion section is required fundamental reconsideration. More information may be added to the discussion of SARA and acute acidosis. Sub-sections might be unnecessary.

5. Conclusion: I'm simply curious whether it confuses readers because the author focuses on SARA yet concludes with acute acidosis.

6. The references section should be written in accordance with journal formatting instructions.

Author Response

Reply to Reviewer Report  (Reviewer-2)

Comment: 1. ABSTRACT: The author has to be improved and this section could be concise.

Response: Thank you very much for your comments and suggestion on our manuscript. These comments are very helpful for revising and improving our paper. According to the reviewer detailed suggestions, abstract has been improved and concise and also added the simple summary and highlighted.

Comment: Introduction part, this research will contribute to the compilation of scientific information on the measuring of Autophagy activity in laminar tissue of laminitis in dairy cows. The introduction section was almost thoroughly re-concisely detailed by focused on laminitis in dairy cows.

Response: Respected reviewer, we have made a revision in introduction section on our manuscript and highlighted in the manuscript.

Comment: 3. Material and methods part, the experiment was conducted to evaluate damage of laminar tissue. - I am concerned about the lack of information on the collection laminar tissue previous induces by oligofructose. - Why does the author use 17g/kg OF to define acute in dairy cows when neither the ruminal pH nor the referee method was measured? - Why didn't the author use an animal without OF as a negative control?

Response: i. Respected reviewer, in our manuscript, according to Thoefner et al. [11]. [Thoefner, M.B.; Wattle, O.; Pollitt, C.C.; French, K.R.; Nielsen, S.S. Histopathology of oligofructose-induced acute laminitis in heifers. J. Dairy Sci. 2005, 88, 2774-2782], the procedure for sampling of hoof laminar tissue was explained.

  1. According to the [Thoefner, et al. Acute bovine laminitis: A new induction model using alimentary oligofructose overload. J. Dairy Sci. 2004, 87, 2932-2940]; [Danscher, et al.. Oligofructose overload induces lameness in cattle. J. Dairy Sci. 2009, 92, 607-616], the 17g/kg OF was suitable amount to induce laminitis in dairy cows as previously used in Danscher et al. and Thoefner et al. research. So we use the same protocol method. The pH, and other clinical parameters are observed in our research model groups as previously published articles (ie,https://doi.org/10.3389/fvets.2020.00351), (https://doi: 10.3389/fvets.2020.597827), and (doi: dx.doi.org/10.21521/mw.6398).

iii. This research model was basically designed and performed to investigate the effect of externally induced OF on dairy cows laminitis while the results of OF induced were compared with conventionally control cows. The control group cows were not induced with OF. So we didn’t use an animal as a negative control.

Comment: 4. Discussion section is required fundamental reconsideration. More information may be added to the discussion of SARA and acute acidosis. Sub-sections might be unnecessary. The suggested change has been incorporated accordingly.

Response: Dear reviewer, thank you, we have made suggested changes about to SARA and acute acidosis has been incorporated in the discussion section on the main manuscript.

Comment: 5. Conclusion: I'm simply curious whether it confuses readers because the author focuses on SARA yet concludes with acute acidosis.

Response: Oligosaccharide (OF) overload in dairy cows has been extensively studied in acute rumen acidosis (ARA), laminitis and synovitis. However, bovine laminitis is secondary to these diseases (septic pleuropneumonia, metritis, mastitis, ruminal acidosis). Therefore, we have made a revision in conclusion section.

Comment: 6. The references section should be written in accordance with journal formatting instructions.

Response: Respected reviewer, we have made a careful revision in the reference section in accordance with journal formatting instructions on the main manuscript.

Reviewer 3 Report

General comments:

The present study aimed to evaluate laminar tissue autophagy status following induced laminitis in dairy cows. This molecular approach represents an advance on pathogeny of non-infectious laminitis development involving autophagy mechanisms. The study is well designed and the results are clearly presented. The comparison of gene expression of ATG12, ATG5, Becilin1, P62 and mTOR on laminar tissue between OF-induced laminitis group and control group, and their differences, show scientific soundness. The conclusions are supported by the results. The references are adequate regarding the autophagy process. Nonetheless, the discussion section is mainly based on several information provided by published studies, and only at final (L430-443) the findings of this study are discussed. I understand that exist a gap on scientific literature addressing this specific subject, which turn hard to make a full discussion; but I suggest to reformulate the discussion section starting by the discussion of the results followed by the respective elucidation of processes.

Specific comments:

L94-95: This hypothesis was previously addressed in a study of your working group (ie, https://doi.org/10.3389/fvets.2020.00351) using the same experimental groups.

L188, 267: Please rename to Table 2 and Table 3, respectively.

L317 (Fig. 2): Please resize the graphics.

L335 (Fig. 3): Please resize the graphic. The “**” is not required in the right bar.  Also, the P-values above the bars are not required if you use the conventional symbols, or vice-versa (report the legend of these symbols). Please check all figures for this issue.

L336 “…diffuse aseptic inflammation…”

L336-381: This paragraph is really necessary?

L382-388: This information should be placed in M&M.

L430-443: The discussion of the results starts here (at the end of the discussion). I suggest to move this part up.

L445-447: Please see L382-388. This model was previously established.

Author Response

Reply to Review report (Reviewer-3)

General comments: The present study aimed to evaluate laminar tissue autophagy status following induced laminitis in dairy cows. This molecular approach represents an advance on pathogeny of non-infectious laminitis development involving autophagy mechanisms. The study is well designed and the results are clearly presented. The comparison of gene expression of ATG12, ATG5, Becilin1, P62 and mTOR on laminar tissue between OF-induced laminitis group and control group, and their differences, show scientific soundness. The conclusions are supported by the results. The references are adequate regarding the autophagy process. Nonetheless, the discussion section is mainly based on several information provided by published studies, and only at final (L430-443) the findings of this study are discussed. I understand that exist a gap on scientific literature addressing this specific subject, which turn hard to make a full discussion; but I suggest to reformulate the discussion section starting by the discussion of the results followed by the respective elucidation of processes.

Response: Thank you very much for your comments and suggestion on our manuscript. These comments are very helpful for revising and improving our paper, as well as the important guiding significance to other research. We have studied the comments carefully and made corrections which we hope meet with approval.

Comment: L94-95: This hypothesis was previously addressed in a study of your working group (ie, https://doi.org/10.3389/fvets.2020.00351) using the same experimental groups.

Response: Yes, Using the same experimental groups (ie,https://doi.org/10.3389/fvets.2020.00351), this study (Autophagy status) was my part of study in a same research group. In the same research group, first author of this study participated equally in all data collection and treatments.

Comment: L188, 267: Please rename to Table 2 and Table 3, respectively.

Response: Respected reviewer, Table 2 and Table 3 has been renamed in the manuscript.

Comment: L317 (Fig. 2): Please resize the graphics.

Response: Respected reviewer, resize the graphic has been corrected in our manuscript.

Comment: L335 (Fig. 3): Please resize the graphic. The “**” is not required in the right bar.  Also, the P-values above the bars are not required if you use the conventional symbols, or vice-versa (report the legend of these symbols). Please check all figures for this issue.

Response: Thank you for your suggestion, we have corrected the p-value and “** in the graphs as your suggestion.

Comment: L336 “…diffuse aseptic inflammation…” L336-381: This paragraph is really necessary? L382-388: This information should be placed in M&M.

Response: In this part ----diffuse aseptic inflammation—at the start of discussion, this paragraph is not necessary. So as your kind suggestion, we have removed this paragraph.

Comment: L430-443: The discussion of the results starts here (at the end of the discussion). I suggest to move this part up.

Response: Dear reviewer, thank you for your suggestion, we have moved the discussion of results to up as your recommendation on the main manuscript.

Comment: L445-447: Please see L382-388. This model was previously established.

Response: Yes, This model was previously established (ie,https://doi.org/10.3389/fvets.2020.00351), Using the same experimental groups, this study (Autophagy status in laminar tissue) was my part of study in a same research group of that model. In the same research group, first author of this study participated equally in all data collection and treatments.

Round 2

Reviewer 1 Report

Dear authors,

the paper has improved seriously, but I still have some small remarks 

line 64: cows

line 67: pleuropneumonia is not a good example, please replace that by mastitis.

Figures: In the figures added to a paper (Fig 1,2,3 here) all abbreviations should be explained in the text beneath the figure, where a Figure can be read independent the text. This essential for me.

line 440+ 444: I prefer the word results for findings 

Author Response

Reviewer-1 comments (2nd Round Revision)

Manuscript ID: Animals-2493363

Dear reviewer,

Thank you very much for giving us an opportunity to again revise our manuscript. We appreciate the reviewer very much for their constructive comments and suggestions on our manuscript entitled “Enhanced Autophagy Activity in Laminar Tissue Damage of Acute Laminitis in Dairy Cows Induced by Oligofructose Overload” (Animals-2493363).

We have studied reviewer’s comments carefully. According to the reviewer’s detailed suggestions, we have made a careful revision on the original manuscript. All revised portions are highlighted in the revised manuscript which we would like to submit for your kind consideration.

Kind regards!

*Corresponding author: Prof. Hongbin Wang

Chairman at Department of Veterinary Surgery, College of Veterinary Medicine, Northeast Agricultural University, Harbin 150030, P. R. China and Director at Heilongjiang Key Laboratory for Laboratory Animals and Comparative Medicine, Harbin 150030, P. R. China

E-mail: hbwang1940@neau.edu.cn ; Tel.:+86-451-55190470

 Reviewer-1

Dear authors,

the paper has improved seriously, but I still have some small remarks 

Response: Respected reviewer, thanks for your comment. I have incorporated accordingly.

line 64: cows

Response: It has been changed as suggested.

line 67: pleuropneumonia is not a good example, please replace that by mastitis.

Response: It has been changed as suggested.

Figures: In the figures added to a paper (Fig 1,2,3 here) all abbreviations should be explained in the text beneath the figure, where a Figure can be read independent the text. This essential for me.

Response: Thanks for your valuable suggestions. The abbreviations have been changed as suggested.

line 440+ 444: I prefer the word results for findings 

Response: Thanks for your valuable suggestions. It has been corrected.
